# Simple Spectral Graph Convolution

**Hao Zhu, Piotr Koniusz**[*]
Australian National University
Canberra, Australia
{hao.zhu,piotr.koniusz}@anu.edu.au

Data61/CSIRO
Canberra, Australia

## Abstract

Graph Convolutional Networks (GCNs) are leading methods for learning graph representations. However, without specially designed architectures, the performance of GCNs degrades quickly with increased depth. As the aggregated neighborhood size and neural network depth are two completely orthogonal aspects of graph representation, several methods focus on summarizing the neighborhood by aggregating $K$-hop neighborhoods of nodes while using shallow neural networks. However, these methods still encounter oversmoothing, and suffer from high computation and storage costs. In this paper, we use a modified Markov Diffusion Kernel to derive a variant of GCN called Simple Spectral Graph Convolution (S$^2$GC). Our spectral analysis shows that our simple spectral graph convolution used in S$^2$GC is a trade-off of low- and high-pass filter bands which capture the global and local contexts of each node. We provide two theoretical claims which demonstrate that we can aggregate over a sequence of increasingly larger neighborhoods compared to competitors while limiting severe oversmoothing. Our experimental evaluations show that S$^2$GC with a linear learner is competitive in text and node classification tasks. Moreover, S$^2$GC is comparable to other state-of-the-art methods for node clustering and community prediction tasks.

## 1 Introduction

In the past decade, deep learning has become mainstream in computer vision and machine learning. Although deep learning has been applied for extraction of features on the Euclidean lattice (Euclidean grid-structured data) with great success, the data in many practical scenarios lies on non-Euclidean structures, whose processing poses a challenge for deep learning. By defining a convolution operator between the graph and signal, Graph Convolutional Networks (GCNs) generalize Convolutional Neural Networks (CNNs) to graph-structured inputs which contain attributes. Message Passing Neural Networks (MPNNs) (Gilmer et al., 2017) unify the graph convolution as two functions: the transformation function and the aggregation function. MPNN iteratively propagates node features based on the adjacency of the graph in a number of rounds.

Despite their enormous success in many applications like social media, traffic analysis, biology, recommendation systems and even computer vision, many of the current GCN models use fairly shallow setting as many of the recent models such as GCN (Kipf & Welling, 2016) achieve their best performance given 2 layers. In other words, 2-layer GCN models aggregate nodes in two-hops neighborhood and thus have no ability to extract information in $K$-hops neighborhoods for $K > 2$. Moreover, stacking more layers and adding a non-linearity tend to degrade the performance of these models. Such a phenomenon is called oversmoothing (Li et al., 2018a), characterized by the effect that as the number of layers increases, the representations of the nodes in GCNs tend to converge to a similar, non-distinctive from one another value. Even adding residual connections, an effective trick for training very deep CNNs, merely slows down the oversmoothing issue (Kipf & Welling, 2016) in GCNs. It appears that deep GCN models gain nothing but the performance degradation from the deep architecture.

One solution for that is to widen the receptive field of aggregation function while limiting the depth of network because the required neighborhood size and neural network depth can be regarded as

---

[*]The corresponding author. The code is available at https://github.com/allenhaozhu/SSGC.

two separate aspects of design. To this end, SGC (Wu et al., 2019) captures the context from $K$-hops neighbours in the graph by applying the $K$-th power of the normalized adjacency matrix in a single layer of neural network. This scheme is also used for attributed graph clustering (Zhang et al., 2019). However, SGC also suffers from oversmoothing as $K \to \infty$, as shown in Theorem 1. PPNP and APPNP (Klicpera et al., 2019a) replace the power of the normalized adjacency matrix with the Personalized PageRank matrix to solve the oversmoothing problem. Although APPNP relieves the oversmoothing problem, it employs a non-linear operation which requires costly computation of the derivative of the filter due to the non-linearity over the multiplication of feature matrix with learnable weights. In contrast, we show that our approach enjoys a free derivative computed in the feed-forward step due to the use of a linear model. Furthermore, APPNP aggregates over multiple $k$-hop neighborhoods ($k = 0, \cdots, K$) but the weighting scheme favors either global or local context making it difficult if not impossible to find a good value of balancing parameter. In contrast, our approach aggregates over $k$-hop neighborhoods in a well-balanced manner.

GDC (Klicpera et al., 2019b) further extends APPNP by generalizing Personalized PageRank (Page et al., 1999) to an arbitrary graph diffusion process. GDC has more expressive power than SGC (Wu et al., 2019), PPNP and APPNP (Klicpera et al., 2019a) but it leads to a dense transition matrix which makes the computation and space storage intractable for large graphs, although authors suggest that the shrinkage method can be used to sparsify the generated transition matrix. Noteworthy are also orthogonal research directions of Sun et al. (2019); Koniusz & Zhang (2020); Elinas et al. (2020) which improve the performance of GCNs by the perturbation of graph, high-order aggregation of features, and the variational inference, respectively.

To tackle the above issues, we propose a Simple Spectral Graph Convolution (S$^2$GC) network for node clustering and node classification in semi-supervised and unsupervised settings. By analyzing the Markov Diffusion Kernel (Fouss et al., 2012), we obtain a very simple and effective spectral filter: we aggregate $k$-step diffusion matrices over $k = 0, \cdots, K$ steps, which is equivalent to aggregating over neighborhoods of gradually increasing sizes. Moreover, we show that our design incorporates larger neighborhoods compared to SGC and copes better with oversmoothing. We explain that limiting overdominance of the largest neighborhoods in the aggregation step limits oversmoothing while preserving the large context of each node. We also show via the spectral analysis that S$^2$GC is a trade-off between the low- and high-pass filter bands which leads to capturing the global and local contexts of each node. Moreover, we show how S$^2$GC and APPNP (Klicpera et al., 2019a) are related and explain why S$^2$GC captures a range of neighborhoods better than APPNP. Our experimental results include node clustering, unsupervised and semi-supervised node classification, node property prediction and supervised text classification. We show that S$^2$GC is highly competitive, often significantly outperforming state-of-the-art methods.

## 2 PRELIMINARIES

**Notations.** Let $G = (V, E)$ be a simple and connected undirected graph with $n$ nodes and $m$ edges. We use $\{1, \cdots, n\}$ to denote the node index of $G$, whereas $d_j$ denotes the degree of node $j$ in $G$. Let $\mathbf{A}$ be the adjacency matrix and $\mathbf{D}$ be the diagonal degree matrix. Let $\widetilde{\mathbf{A}} = \mathbf{A} + \mathbf{I}_n$ denote the adjacency matrix with added self-loops and the corresponding diagonal degree matrix $\widetilde{\mathbf{D}}$, where $\mathbf{I}_n \in \mathcal{S}_{++}^n$ is an identity matrix. Finally, let $\mathbf{X} \in \mathbb{R}^{n \times d}$ denote the node feature matrix, where each node $v$ is associated with a $d$-dimensional feature vector $\mathbf{X}_v$. The normalized graph Laplacian matrix is defined as $\mathbf{L} = \mathbf{I}_n - \mathbf{D}^{-1/2}\mathbf{A}\mathbf{D}^{-1/2} \in \mathcal{S}_+^n$, that is, a symmetric positive semidefinite matrix with eigendecomposition $\mathbf{U}\mathbf{\Lambda}\mathbf{U}^\top$, where $\mathbf{\Lambda}$ is a diagonal matrix with eigenvalues of $\mathbf{L}$, and $\mathbf{U} \in \mathbb{R}^{n \times n}$ is a unitary matrix that consists of the eigenvectors of $\mathbf{L}$.

**Spectral Graph Convolution (Defferrard et al., 2016).** We consider spectral convolutions on graphs defined as the multiplication of signal $\mathbf{x} \in \mathbb{R}^n$ with a filter $g_\theta$ parameterized by $\boldsymbol{\theta} \in \mathbb{R}^n$ in the Fourier domain:

$$g_\theta(\mathbf{L}) * x = \mathbf{U}g_\theta^*(\mathbf{\Lambda})\mathbf{U}^\top \mathbf{x}, \tag{1}$$

where the parameter $\boldsymbol{\theta} \in \mathbb{R}^n$ is a vector of spectral filter coefficients. One can understand $g_\theta$ as a function operating on eigenvalues of $\mathbf{L}$, that is, $g_\theta^*(\mathbf{\Lambda})$. To avoid eigendecomposition, $g_\theta(\mathbf{\Lambda})$ can be approximated by a truncated expansion in terms of Chebyshev polynomials $T_k(\mathbf{\Lambda})$ up to the $K$-th

order (Defferrard et al., 2016):

$$g_\theta^*(\mathbf{\Lambda}) \approx \sum_{k=0}^{K-1} \theta_k T_k(\tilde{\mathbf{\Lambda}}), \tag{2}$$

with a rescaled $\tilde{\mathbf{\Lambda}} = \frac{1}{2\lambda_{\max}}\mathbf{\Lambda} - \mathbf{I}_n$, where $\lambda_{\max}$ denotes the largest eigenvalue of $\mathbf{L}$ and $\boldsymbol{\theta} \in \mathbb{R}^K$ is now a vector of Chebyshev coefficients.

**Vanila Graph Convolutional Network (GCN) (Kipf & Welling, 2016).** The vanilla GCN is a first-order approximation of spectral graph convolutions. If one sets $K = 1$, $\theta_0 = 2$, and $\theta_1 = -1$ for Eq. 2, they obtain the convolution operation $g_\theta(\mathbf{L}) * \mathbf{x} = (\mathbf{I} + \mathbf{D}^{-1/2}\mathbf{A}\mathbf{D}^{-1/2})\mathbf{x}$. Finally, by the renormalization trick, replacing matrix $\mathbf{I} + \mathbf{D}^{-1/2}\mathbf{A}\mathbf{D}^{-1/2}$ by a normalized version $\widetilde{\mathbf{T}} = \widetilde{\mathbf{D}}^{-1/2}\widetilde{\mathbf{A}}\widetilde{\mathbf{D}}^{-1/2} = (\mathbf{D} + \mathbf{I}_n)^{-1/2}(\mathbf{A} + \mathbf{I}_n)(\mathbf{D} + \mathbf{I}_n)^{-1/2}$ leads to the GCN layer with a non-linear function $\sigma$:

$$H^{(l+1)} = \sigma(\widetilde{\mathbf{T}}\mathbf{H}^{(l)}\mathbf{W}^{(l)}). \tag{3}$$

**Graph Diffusion Convolution (GDC) (Klicpera et al., 2019b).** A generalized graph diffusion is given by the diffusion matrix:

$$\mathbf{S} = \sum_{k=0}^{\infty} \theta_k \mathbf{T}^k, \tag{4}$$

with the weighting coefficients $\theta_k$ and the generalized transition matrix $\mathbf{T}$. Eq. 4 can be regarded as related to the Taylor expansion of matrix-valued functions. Thus, the choice of $\theta_k$ and $\mathbf{T}^k$ must at least ensure that Eq. 4 converges. Klicpera et al. (2019b) provide two special cases as low-pass filters *ie.*, the heat kernel and the kernel based on random walk with restarts. If $\mathbf{S}$ denotes the adjacency matrix and $\mathbf{D}$ is the diagonal degree matrix of $\mathbf{S}$, the corresponding graph diffusion convolution is then defined as $\mathbf{D}^{-1/2}\mathbf{S}\mathbf{D}^{-1/2}\mathbf{x}$. Note that $\theta_k$ can be a learnable parameter, or it can be chosen in some other way. Many works use the expansion in Eq. 4 but different choices of $\theta_k$ realise very different filters, making each method unique.

**Simple Graph Convolution (SGC) (Wu et al., 2019).** A classical MPNN (Gilmer et al., 2017) averages (in each layer) the hidden representations among 1-hop neighbors. This implies that each node in the $K$-th layer obtains feature information from all nodes that are $K$-hops away in the graph. By hypothesizing that the non-linearity between GCN layers is not critical, SGC captures information from $K$-hops neighborhood in the graph by applying the $K$-th power of the transition matrix in a single neural network layer. The SGC can be regarded as a special case of GDC without non-linearity and without the normalization by $\mathbf{D}^{-1/2}$ if we set $\theta_K = 1$ and $\theta_{i<K} = 0$ in Eq. 4, and $\mathbf{T} = \tilde{\mathbf{T}}$, which yields:

$$\hat{Y} = \mathrm{softmax}(\widetilde{\mathbf{T}}^K \mathbf{X}\mathbf{W}). \tag{5}$$

Although SGC is an efficient and effective method, increasing $K$ leads to oversmoothing. Thus, SGC uses a small $K$ number of layers. Theorem 1 shows that oversmoothing is a result of convergence to the stationary distribution in the graph diffusion process when time $t \to \infty$.

**Theorem 1.** *(Chung & Graham, 1997) Let $\lambda_2$ denote the second largest eigenvalue of transition matrix $\widetilde{\mathbf{T}} = \mathbf{D}^{-1}\mathbf{A}$ of a non-bipartite graph, $\mathbf{p}(t)$ be the probability distribution vector and $\boldsymbol{\pi}$ the stationary distribution. If walk starts from the vertex $i$, $p_i(0) = 1$, then after $t$ steps for every vertex, we have:*

$$|p_j(t) - \pi_j| \leq \sqrt{\frac{d_j}{d_i}} \lambda_2^t. \tag{6}$$

**APPNP.** Klicpera et al. (2019a) proposed to use the Personalized PageRank to derive a fixed filter of order $K$. Let $f_\theta(\mathbf{X})$ denote the output of a two-layer fully connected neural network on the feature matrix $\mathbf{X}$, then the PPNP model is defined as $\mathbf{H} = \alpha\mathbf{I}_n - (1 - \alpha)\widetilde{\mathbf{T}}^{-1}f_\theta(\mathbf{X})$. To avoid calculating the inverse of matrix $\widetilde{\mathbf{T}}$, Klicpera et al. (2019a) also propose the Approximate PPNP (APPNP), which replaces the costly inverse with an approximation by the truncated power iteration:

$$\mathbf{H}^{(l+1)} = (1 - \alpha)\widetilde{\mathbf{T}}\mathbf{H}^{(l)} + \alpha\mathbf{H}^{(0)}, \tag{7}$$

where $\mathbf{H}^{(0)} = f_\theta(\mathbf{X}) = \mathrm{ReLU}(\mathbf{X}\mathbf{W})$ or $\mathbf{H}^{(0)} = f_\theta(\mathbf{X}) = \mathrm{MLP}(\mathbf{X})$. By decoupling feature transformation and propagation steps, PPNP and APPNP aggregate information from multi-hop neighbors.

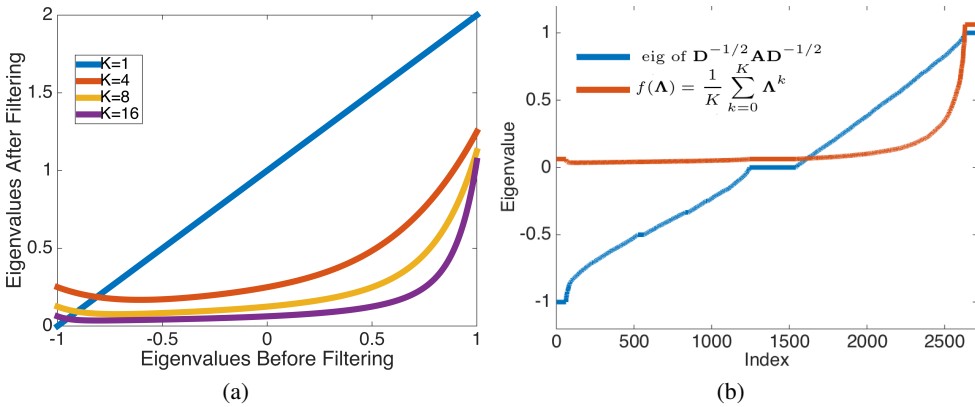

Figure 1: (a) Function $f(\lambda) = \frac{1}{K}\sum_{k=0}^{K}\lambda^k$ with $\lambda \in [-1,1]$, $K \in \{1,4,8,16\}$; (b) Sorted by index, eigenvalues of $\mathbf{D}^{-1/2}\mathbf{A}\mathbf{D}^{-1/2}$ and push-forward eigenvalues $f(\mathbf{\Lambda}) = \frac{1}{K}\sum_{k=0}^{K}\mathbf{\Lambda}^k$ on Cora network ($K = 16$).

## 3 METHODOLOGY

Below, we firstly outline two claims which underlie the design of our network, with the goal of mitigating oversmoothing. Moreover, we analyze the Markov Diffusion Kernel (Fouss et al., 2012) and note that it acts as a low-pass spectral filter of various degree. Based on the feature mapping function underlying this kernel, we present our Simple Spectral Graph Convolution network and discuss its relation with other models. Finally, we provide the comparison of computational and storage complexity requirements.

### 3.1 MOTIVATION

Our design follows Claims I and II described in Section A.3, which includes their detailed proofs.

**Claim I.** By design, our filter gives the highest weight to the closest neighborhood of a node, as neighborhoods $\mathcal{N}$ of diffusion steps $k = 0, \cdots, K$ obey $\mathcal{N}(\widetilde{\mathbf{T}}^0) \subseteq \mathcal{N}(\widetilde{\mathbf{T}}^1) \subseteq \cdots \subseteq \mathcal{N}(\widetilde{\mathbf{T}}^K) \subseteq \mathcal{N}(\widetilde{\mathbf{T}}^\infty)$. That is, smaller neighborhoods belong to larger neighborhoods too.

**Claim II.** As $K \to \infty$, the ratio of energies contributed by S$^2$GC to SGC is 0. Thus, the energy of infinite-dimensional receptive field (largest $k$) will not dominate the sum energy of our filter. Thus, S$^2$GC can incorporate larger receptive fields without undermining contributions of smaller receptive fields. This is substantiated by Table 8, where S$^2$GC achieves the best results for $K = 16$, whereas SGC achieves poorer results by comparison, whose peak is at $K = 4$ (note that larger $K$ is better).

### 3.2 MARKOV DIFFUSION KERNEL

Two nodes are considered similar when they are diffused in a similar way through the graph, as then they influence the other nodes in a similar manner (Fouss et al., 2012). Moreover, two nodes are close neighbors if they are in the same distinct cluster. The Markov Diffusion distance between nodes $i$ and $j$ at time $K$ is defined as:

$$d_{ij}(K) = \|\mathbf{x}_i(K) - \mathbf{x}_j(K)\|_2^2, \tag{8}$$

where the average visiting rate $\mathbf{x}_i(K)$ after $K$ steps for a process that started at time $k = 0$ is computed as follows:

$$\mathbf{x}_i(K) = \frac{1}{K}\sum_{k=1}^{K}\mathbf{T}^k\mathbf{x}_i(0). \tag{9}$$

By defining $\mathbf{Z}(K) = \frac{1}{K}\sum_{k=1}^{K}\mathbf{T}^k$, we reformulate Eq. 8 as the following metric:

$$d_{ij}(K) = \|\mathbf{Z}(K)(\mathbf{x}_i(0) - \mathbf{x}_j(0))\|_2^2. \tag{10}$$

The underlying feature map of Markov Diffusion Kernel (MDK) is given as $\mathbf{Z}(K)\mathbf{x}_i(0)$ for node $i$.

The effect of the linear projection $\mathbf{Z}(K)$ (filter) acting on spectrum as $f(\lambda) = \frac{1}{K} \sum_{k=0}^{K} \lambda^k$ (we sum from 0 to include self-loops) is plotted in Figure 1, from which we observe the following properties: (i) $\mathbf{Z}(K)$ preserves leading (large) eigenvalues of $\mathbf{T}$ and (ii) the higher $K$ is the stricter the low-pass filter becomes but the filter also preserves the high frequency. In other words, as $K$ grows, this filter includes larger and larger neighborhood but also maintains the closest locality of nodes. Note that $\mathbf{L} = \mathbf{I} - \mathbf{T}$, where $\mathbf{L}$ is the normalized Laplacian matrix and $\mathbf{T}$ is the normalized adjacency matrix. Thus, keeping large positive eigenvalues for $\mathbf{T}$ equals keeping small eigenvalues for $\mathbf{L}$.

## 3.3 SIMPLE SPECTRAL GRAPH CONVOLUTION

Based on the aforementioned Markov Diffusion Kernel, we include self-loops and propose the Simple Spectral Graph Convolution ($S^2GC$) network with the softmax classifier after the linear layer:

$$\hat{Y} = \text{softmax}(\frac{1}{K} \sum_{k=0}^{K} \widetilde{\mathbf{T}}^k \mathbf{X} \mathbf{W}). \tag{11}$$

Let $\|\mathbf{x}_i\|_2 = 1, \forall i$ (each $\mathbf{x}_i$ is a row in $\mathbf{X}$). If $K \to \infty$ then $\mathbf{H} = \sum_{k=0}^{\infty} \widetilde{\mathbf{T}}^k \mathbf{X}$ is the optimal diffused representation of the normalized Laplacian Regularization problem given below:

$$\underset{\substack{\mathbf{H} \\ \text{s.t. } \|\mathbf{h}_i\|_2 = 1, \forall i}}{\arg \min} \quad q(\mathbf{H}), \quad \text{where} \quad q(\mathbf{H}) = \frac{1}{2} \bigg( \sum_{i,j=1}^{n} \widetilde{\mathbf{A}}_{ij} \| \frac{\mathbf{h}_i}{\sqrt{d_i}} - \frac{\mathbf{h}_j}{\sqrt{d_j}} \|_2^2 \bigg) + \frac{1}{2} \bigg( \sum_{i=1}^{n} \|\mathbf{h}_i - \mathbf{x}_i\|_2^2 \bigg), \tag{12}$$

and each vector $\mathbf{h}_i$ denotes the $i$-th row of $\mathbf{H}$. Compared with the more common form in (Zhou et al., 2004), we impose $\|\mathbf{h}_i\|_2^2 = \|\mathbf{x}_i\|_2^2 = 1$, to minimize the difference between $\mathbf{h}_i$ and $\mathbf{x}_i$ via the cosine distance rather than the Euclidean distance. Differentiating $q(\mathbf{H})$ with respect to $\mathbf{H}$, we have $\widetilde{\mathbf{L}}\mathbf{H} - \mathbf{X} = 0$. Thus, the optimal representation $\mathbf{H} = (\mathbf{I} - \widetilde{\mathbf{T}})^{-1}\mathbf{X}$, where $(\mathbf{I} - \widetilde{\mathbf{T}})^{-1} = \sum_{k=0}^{\infty} \widetilde{\mathbf{T}}^k$. However, the infinite expansion resulting from Eq. 12 is in fact suboptimal due to oversmoothing. Thus, we include in Eq. 11 a self-loop $\widetilde{\mathbf{T}}^0 = \mathbf{I}$, the $\alpha \in [0, 1]$ parameter (Table 9 evaluates its impact) to balance the self-information of node *vs.* consecutive neighborhoods, and we consider finite $K$. We generalize the Eq. 11 as:

$$\hat{Y} = \text{softmax}\bigg( \frac{1}{K} \sum_{k=1}^{K} \Big( (1 - \alpha) \widetilde{\mathbf{T}}^k \mathbf{X} + \alpha \mathbf{X} \Big) \mathbf{W} \bigg). \tag{13}$$

**Relation of $S^2GC$ to GDC.** GDC uses the entire filter matrix $\mathbf{S}$ of size $n \times n$ as $\mathbf{S}$ is re-normalized numerous times by its degree. Klicpera et al. (2019b) explain that 'most graph diffusions result in a dense matrix $\mathbf{S}$'.

In contrast, our approach is simply computed as $(\sum_{k=1}^{K} \widetilde{\mathbf{T}}^k \mathbf{X})\mathbf{W}$ (plus the self-loop), where $\mathbf{X}$ is of size $n \times d$, and $d \ll n$, where $n$ and $d$ are the number of nodes and features, respectively. The $\widetilde{\mathbf{T}}^k \mathbf{X}$ step is computed as $\widetilde{\mathbf{T}} \cdot (\widetilde{\mathbf{T}} \cdot (\cdots (\widetilde{\mathbf{T}}\mathbf{X}) \cdots))$, which requires $K$ sparse matrix-matrix multiplications between a sparse matrices of size $n \times n$ and a dense matrix of size $n \times d$. Thus, $S^2GC$ can handle extremely large graphs as $S^2GC$ does not need to sparsify dense filter matrices (in contrast to GDC).

**Relation of $S^2GC$ to APPNP.** Let $\mathbf{H}^0 = \mathbf{X}\mathbf{W}$ as we use the linear step in our $S^2GC$. Then and only then, for $l = 0$ and $\mathbf{H}^0 = \mathbf{X}\mathbf{W}$, APPNP expansion yields $\mathbf{H}^1 = (1 - \alpha)\widetilde{\mathbf{T}}\mathbf{X}\mathbf{W} + \alpha\mathbf{X}\mathbf{W} = ((1 - \alpha)\widetilde{\mathbf{T}} + \alpha\mathbf{I})\mathbf{X}\mathbf{W}$, which is equal to our $\mathbf{Z}(1)\mathbf{X}\mathbf{W} = (\sum_{k=0}^{K} \widetilde{\mathbf{T}}^k)\mathbf{X}\mathbf{W} = \widetilde{\mathbf{T}}\mathbf{X} + \mathbf{X} = (\widetilde{\mathbf{T}} + \mathbf{I})\mathbf{X}\mathbf{W}$ if $\alpha = 0.5$, $K = 1$, except for scaling (constant) of $\mathbf{H}^1$.

In contrast, for $l = 1$ and general case $\mathbf{H}^0 = f(\mathbf{X}; \mathbf{W})$, APPNP yields $\mathbf{H}^2 = (1 - \alpha)^2 \widetilde{\mathbf{T}}^2 f(\mathbf{X}; \mathbf{W}) + (1 - \alpha)\alpha\widetilde{\mathbf{T}}f(\mathbf{X}; \mathbf{W}) + \alpha f(\mathbf{X}; \mathbf{W})$ from which it is easy to note specific weight coefficients $(1 - \alpha)^2$, $(1 - \alpha)\alpha$ and $\alpha$ associated with 2-, 1-, and 0-hops. This shows that the APPNP expansion is very different to the $S^2GC$ expansion in Eq. 13. In fact, $S^2GC$ and APPNP are only equivalent if $\alpha = 0.5$, $K = 1$ and $f$ is the linear transformation.

Table 1: Computational and storage complexities $\mathcal{O}(\cdot)$.

| Stage | Complexity | APPNP | GDC | SGC | S$^2$GC |
|---|---|---|---|---|---|
| Forward | Computation Cost | $K|E|d + Knd$ | $\approx K|E|n$ | $K|E|d$ | $K|E|d + Knd$ |
| Propagation | Storage Cost | $nd + |E|$ | $\approx n^2$ | $nd + |E|$ | $nd + |E|$ |
| Backward | Computation Cost | $K|E|d$ | 0 | 0 | 0 |
| Propagation | Storage Cost | $nd + |E|$ | 0 | 0 | 0 |

Moreover, as APPNP assumes $\mathbf{H}^0 = f(\mathbf{X}; \mathbf{W}) = \text{ReLU}(\mathbf{XW})$ (or MLP in place of ReLU), their optimizer has to backpropagate through $f(\mathbf{X}; \mathbf{W})$ to obtain $\frac{\partial f}{\partial \mathbf{W}}$ and multiply this result with the above expansion $e.g.$, $\frac{\partial \mathbf{H}^2}{\partial \mathbf{W}} = (1-\alpha)^2 \widetilde{\mathbf{T}}^2 f'(\mathbf{X}; \mathbf{W}) + (1-\alpha)\alpha \widetilde{\mathbf{T}} f'(\mathbf{X}; \mathbf{W}) + \alpha f'(\mathbf{X}; \mathbf{W})$.

In contrast, we use the linear function $\mathbf{XW}$. Thus, $\frac{\partial \mathbf{XW}}{\partial \mathbf{W}}$ yields $\mathbf{X}$. Thus, the multiplication of our expansion with $\mathbf{X}$ for the backpropagation step is in fact obtained in the forward pass which makes our approach very fast for large graphs.

**Relation of S$^2$GC to AR.** The AR filter (Li et al., 2019) uses the regularized Laplacian kernel (Smola & Kondor, 2003) which differs from the (modified) Markov Diffusion Kernel used by us. Specifically, the regularized Laplacian kernel uses the negated Laplacian matrix $-\mathbf{L}$ yielding $\mathbf{K}_\mathrm{L} = \sum_{k=0}^{\infty} \alpha^k (-\mathbf{L})^k = (\mathbf{I} + \alpha\mathbf{L})^{-1}$, where $\mathbf{L} = \mathbf{I} - \widetilde{\mathbf{T}}$, which is related to the von Neumann diffusion kernel $\mathbf{K}_\mathrm{vN} = \sum_{k=0}^{\infty} \alpha^k \mathbf{A}^k$. In contrast, the Markov Diffusion Kernel is defined as $\mathbf{K}_\mathrm{MD}(K) = \mathbf{Z}(K)\mathbf{Z}^\mathrm{T}(K)$, where $\mathbf{Z}(K) = \frac{1}{K} \sum_{k=1}^{K} \widetilde{\mathbf{T}}^k$ and $\widetilde{\mathbf{T}} = \mathbf{D}^{-1/2} \mathbf{A} \mathbf{D}^{-1/2}$.

**Relation of S$^2$GC to Jumping Knowledge Network (JKN).** Xu et al. (2018b) combine intermediate node representations from each layer by concatenating them in the final layer. However, (Xu et al., 2018b) use non-linear layers, which results in a completely different network architecture and the usual slower processing time due to the complex backpropagation chain.

### 3.4 COMPLEXITY ANALYSIS

For S$^2$GC, the storage costs is $\mathcal{O}(|E| + nd)$, where $|E|$ is the total edge count, $nd$ relates to saving the $\widetilde{\mathbf{T}}^k\mathbf{X}$ during intermediate multiplications $\widetilde{\mathbf{T}} \cdot (\widetilde{\mathbf{T}} \cdot (\cdots (\widetilde{\mathbf{T}}\mathbf{X}) \cdots))$. The computational cost is $\mathcal{O}(K|E|d + Knd)$. Each sparse matrix-matrix multiplication $\widetilde{\mathbf{T}}\mathbf{X}$ costs $|E|d$. We need $K$ such multiplications, where $Knd$ and $nd$ are costs of summation over filters and adding features $\mathbf{X}$.

In contrast, the storage cost of GDC is approximately $\mathcal{O}(n^2)$, and the computational cost is approximately $\mathcal{O}(K|E|n)$, where $n$ is the node numbers, $K$ is the order of terms and $|E|$ is the number of graph edges. APPNP, SGC and S$^2$GC have much lower cost than GDC. Note that $K|E|d \gg Knd$ and $n \gg d$. We found that APPNP, SGC and S$^2$GC have similar computational and storage costs in the forward stage. We note that symbol $d$ in APPNP is not the dimension of features $\mathbf{X}$ but dimension of $f(\mathbf{X})$, which is the number of categories.

For the backward stage including computations of the gradient of the classification step, the computational costs of GDC, SGC and S$^2$GC are independent of $K$ and $|E|$ because the graph convolution for these methods does not require backpropagation (the gradients is computed in the forward step). In contrast, APPNP requires backprop as explained earlier.

Table 1 summarizes the computational and storage costs of several methods. Table 2 demonstrates that APPNP is over $66\times$ slower than S$^2$GC on the large scale Products dataset (OGB benchmark) despite, for fairness, we use the same basic building blocks of PyTorch among compared methods.

## 4 EXPERIMENTS

In this section, we evaluate the proposed method on four different tasks: node clustering, community prediction, semi-supervised node classification and text classification.

Table 2: Timing (seconds) on Cora, Citeseer, Pubmed and the large scale Open Graph Benchmark (OGB) which includes Products.

| methods | Cora | Citeseer | Pubmed | Products |
|---------|------|----------|--------|----------|
| SGC | 0.45 | 0.55 | 0.78 | 9.8 |
| APPNP | 1.08 | 1.44 | 1.32 | 748 |
| $S^2GC$ | 0.67 | 0.81 | 0.79 | 11.4 |

Table 3: Clustering performance with three different metrics on four datasets.

| Methods | Input | Cora | | | Citeseer | | | Pubmed | | | Wiki | | |
|---------|-------|------|------|------|----------|------|------|--------|------|------|------|------|------|
| | | Acc% | NMI% | F1% | Acc% | NMI% | F1% | Acc% | NMI% | F1% | Acc% | NMI% | F1% |
| k-means | Feature | 34.65 | 16.73 | 25.42 | 38.49 | 17.02 | 30.47 | 57.32 | 29.12 | 57.35 | 33.37 | 30.20 | 24.51 |
| Spectral-f | Feature | 36.26 | 15.09 | 25.64 | 46.23 | 21.19 | 33.70 | 59.91 | 32.55 | 58.61 | 41.28 | 43.99 | 25.20 |
| Spectral-g | Graph | 34.19 | 19.49 | 30.17 | 25.91 | 11.84 | 29.48 | 39.74 | 3.46 | 51.97 | 23.58 | 19.28 | 17.21 |
| DeepWalk | Graph | 46.74 | 31.75 | 38.06 | 36.15 | 9.66 | 26.70 | 61.86 | 16.71 | 47.06 | 38.46 | 32.38 | 25.74 |
| GAE | Both | 53.25 | 40.69 | 41.97 | 41.26 | 18.34 | 29.13 | 64.08 | 22.97 | 49.26 | 17.33 | 11.93 | 15.35 |
| VGAE | Both | 55.95 | 38.45 | 41.50 | 44.38 | 22.71 | 31.88 | 65.48 | 25.09 | 50.95 | 28.67 | 30.28 | 20.49 |
| ARGE | Both | 64.00 | 44.90 | 61.90 | 57.30 | 35.00 | 54.60 | 59.12 | 23.17 | 58.41 | 41.40 | 39.50 | 38.27 |
| ARVGE | Both | 62.66 | 45.28 | 62.15 | 54.40 | 26.10 | 52.90 | 58.22 | 20.62 | 23.04 | 41.55 | 40.01 | 37.80 |
| AGC | Both | 68.92 | 53.68 | 65.61 | 67.00 | 41.13 | 62.48 | 69.78 | 31.59 | 68.72 | 47.65 | 45.28 | 40.36 |
| $S^2GC$ | Both | **69.60** | **54.71** | **65.83** | **69.11** | **42.87** | **64.65** | **70.98** | **33.21** | **70.28** | **52.67** | **49.62** | **44.31** |

## 4.1 NODE CLUSTERING

We compare $S^2GC$ with three variants of clustering: (i) Methods that only use node features *ie.*, k-means and spectral clustering (spectral-f) that construct a similarity matrix with the node features by a linear kernel. (ii) Structural clustering methods that only use graph structures *ie.*, spectral clustering (spectral-g) that takes the node adjacency matrix as the similarity matrix, DeepWalk (Perozzi et al., 2014), and (iii) Attributed graph clustering methods that utilize both node features and graph structures: Graph Autoencoder (GAE) and Graph Variational Autoencoder (VGAE) (Kipf & Welling, 2016), and Adversarially Regularized Graph Autoencoder (ARGE), Variational Graph Autoencoder (ARVGE) (Pan et al., 2018) and AGC (Zhang et al., 2019). To evaluate the clustering performance, three performance measures are adopted: clustering Accuracy (Acc), Normalized Mutual Information (NMI) and macro F1-score (F1). We run each method 10 times on four datasets: Cora, CiteSeer, PubMed, and Wiki, and we report the average clustering results in Table 3, where top-1 results are highlighted in bold. To adaptively select the order $K$, we use the clustering performance metric: internal criteria based on the information intrinsic to the data alone Zhang et al. (2019).

## 4.2 COMMUNITY PREDICTION

We supplement our social network analysis by using $S^2GC$ to inductively predict the community structure on Reddit, a large scale dataset, as shown in Table 10, which cannot be processed by the vanilla GCN Kipf & Welling (2016) and GDC (Klicpera et al., 2019b) due to the memory issues. On the Reddit dataset, we train $S^2GC$ with L-BFGS using no regularization, and we set $K = 5$ and $\alpha = 0.05$. We evaluate $S^2GC$ inductively according to protocol (Chen et al., 2018). We train $S^2GC$ on a subgraph comprising only training nodes and test on the original graph. On all datasets, we tune the number of epochs based on both the convergence behavior and the obtained validation accuracy.

For Reddit, we compare $S^2GC$ to the reported performance of supervised and unsupervised variants of GraphSAGE (Hamilton et al., 2017), FastGCN (Chen et al., 2018), SGC (Wu et al., 2019) and DGI (Velickovic et al., 2019). Table 4 also highlights the setting of the feature extraction step for each method. Note that $S^2GC$ and SGC involve no learning because they do not learn any parameters to extract features. The logistic regression is used as the classifier for both unsupervised and no-learning approaches to train with labels afterward.

## 4.3 NODE CLASSIFICATION

For the semi-supervised node classification task, we apply the standard fixed training, validation and testing splits (Yang et al., 2016) on the Cora, Citeseer, and Pubmed datasets, with 20 nodes per class for training, 500 nodes for validation and 1,000 nodes for testing. For baselines, We include

Table 4: Test Micro F1 Score (%) averaged over 10 runs on Reddit. Results of other models are taken from their papers.

| Setting | Model | Test F1 |
|---|---|---|
| Supervised | SAGE-mean | 95.0 |
| | SAGE-LSTM | **95.4** |
| Unsupervised | SAGE-GCN | 93.0 |
| | FastGCN | 93.7 |
| | SAGE-mean | 89.7 |
| | SAGE-LSTM | 90.7 |
| No Learning | SAGE-GCN | 90.8 |
| | DGI | 94.0 |
| | SGC | 94.9 |
| | S$^2$GC | **95.3** |

Table 5: Test accuracy (%) averaged over 10 runs on citation networks.

| methods | Cora | Citeseer | Pubmed |
|---|---|---|---|
| GCN | 81.4± 0.4 | 70.9± 0.5 | 79.0± 0.4 |
| GAT | 83.3± 0.7 | 72.6± 0.6 | 78.5± 0.3 |
| FastGCN | 79.8± 0.3 | 68.8± 0.6 | 77.4± 0.3 |
| GIN | 77.6± 1.1 | 66.1± 0.9 | 77.0± 1.2 |
| DGI | 82.5± 0.7 | 71.6± 0.7 | 78.4± 0.7 |
| SGC | 81.0± 0.03 | 71.9± 0.11 | 78.9± 0.01 |
| MixHop | 81.8±0.6 | 71.4±0.8 | 80.0±1.1 |
| APPNP | 83.3±0.5 | 71.7±0.6 | 80.1±0.2 |
| Chebynet | 78.0± 0.4 | 70.1± 0.5 | 78.0± 0.4 |
| AR filter | 80.8± 0.02 | 69.3± 0.15 | 78.1± 0.01 |
| Ours | **83.5± 0.02** | **73.6± 0.09** | **80.2± 0.02** |

Table 6: Test accuracy (%) averaged over 10 runs on the large-scale OGB node property prediction benchmark.

| methods | Products | Mag | Arxiv |
|---|---|---|---|
| MLP | 61.06±0.08 | 26.92±0.26 | 55.50±0.23 |
| GCN | 75.64±0.21 | 30.43±0.25 | 71.74±0.29 |
| GraphSage | **78.29±0.16** | 31.53±0.15 | 71.49±0.27 |
| Softmax | 47.70±0.03 | 24.13±0.03 | 52.77±0.56 |
| SGC | 68.87± 0.01 | 29.47±0.03 | 68.78±0.02 |
| S$^2$GC | 70.22± 0.01 | 32.47±0.11 | 70.15±0.13 |
| S$^2$GC+MLP | 76.84±0.20 | **32.72±0.23** | **72.01±0.25** |

three state-of-the-art shallow models: GCN (Kipf & Welling, 2016), GAT (Veličković et al., 2017), FastGCN (Chen et al., 2018), APPNP (Klicpera et al., 2019a), Mixhop (Abu-El-Haija et al., 2019), SGC (Wu et al., 2019), DGI (Velickovic et al., 2019) and GIN (Xu et al., 2018a). We use the Adam SGD optimizer (Kingma & Ba, 2014) with a learning rate of 0.02 to train S$^2$GC. We set $\alpha = 0.05$ and $K = 16$ on all datasets. To determine $K$ and $\alpha$, we used the MetaOpt package Bergstra et al. (2015) with 20 steps to meta-optimize hyperparameters on the validation set of Cora. Following that, we fixed $K = 16$ and $\alpha = 0.05$ across all datasets so $K$ and $\alpha$ are not tuned to individual datasets at all. We will discuss the influence of $\alpha$ and $K$ later.

To evaluate the proposed method on large scale benchmarks (see Table 6), we use Arxiv, Mag and Products datasets to compare the proposed method with SGC, GraphSage, GCN, MLP and Softmax (multinomial Regression). On these three datasets, our method consistently outperforms SGC. On Arxiv and Products, our method cannot outperform GCN and GraphSage while MLP outperforms softmax classifier significantly. Thus, we argue that MLP plays a more important role here than the graph convolution. To prove this point, we also conduct an experiment (S$^2$GC+MLP) for which we use MLP in place of the linear classifier, and we obtain a more powerful variant of S$^2$GC. On Mag, S$^2$GC+MLP outperforms S$^2$GC by a tiny margin because the performance of MLP is close to the one of softmax. On other two datasets, S$^2$GC+MLP is a very strong performer. Our S$^2$GC+MLP is the best performer on Mag and Arxiv.

## 4.4 TEXT CLASSIFICATION

Text classification predicts the labels of documents. Yao et al. (2019) use a 2-layer GCN to achieve state-of-the-art results by creating a corpus-level graph, which treats both documents and words as nodes in a graph. Word-to-word edge weights are given by Point-wise Mutual Information (PMI) and word-document edge weights are given by the normalized TF-IDF scores.

We ran our experiments on five widely used benchmark corpora including the Movie Review (MR), 20-Newsgroups (20NG), Ohsumed, R52 and R8 of Reuters 21578. We first preprocessed all datasets by cleaning and tokenizing text as Kim (2014). We then removed stop words defined in NLTK6 and low-frequency words appearing less than 5 times for 20NG, R8, R52 and Ohsumed. We compare our method with GCN (Kipf & Welling, 2016) and SGC (Wu et al., 2019). The statistics of the

Table 7: Test accuracy on the document classification task.

| Model | 20NG | R8 | R52 | Ohsumed | MR |
|-------|------|-----|-----|---------|-----|
| Text GCN | $87.9 \pm 0.2$ | $97.0 \pm 0.2$ | $93.8 \pm 0.2$ | $68.2 \pm 0.4$ | $76.3 \pm 0.3$ |
| SGC | $88.5 \pm 0.1$ | $97.2 \pm 0.2$ | $94.0 \pm 0.2$ | $\mathbf{68.5 \pm 0.3}$ | $75.9 \pm 0.3$ |
| $S^2$GC | $\mathbf{88.6 \pm 0.1}$ | $\mathbf{97.4 \pm 0.1}$ | $\mathbf{94.5 \pm 0.2}$ | $\mathbf{68.5 \pm 0.1}$ | $\mathbf{76.7 \pm 0.0}$ |

Table 8: Summary of classification accuracy (%) w.r.t. various depths. In the linear model, the filter parameter $K$ is equivalent to the number of layers.

| Dataset | Method | Layers (K) | | | | | |
|---------|--------|-----|-----|-----|-----|-----|-----|
| | | 2 | 4 | 8 | 16 | 32 | 64 |
| Cora | GCN | 81.1 | 80.4 | 69.5 | 64.9 | 60.3 | 28.7 |
| | SGC | 80.8 | 81.5 | 80.7 | 79.0 | 75.9 | 66.8 |
| | $S^2$GC | 76.2 | 79.8 | 82.2 | **83.5** | 82.6 | 82.0 |
| Citeseer | GCN | 70.8 | 67.6 | 30.2 | 18.3 | 25.0 | 20.0 |
| | SGC | 71.9 | 72.6 | 73.1 | 72.2 | 70.6 | 69.2 |
| | $S^2$GC | 70.7 | 72.6 | 72.7 | 73.6 | **74.0** | 73.4 |
| Pubmed | GCN | 79.0 | 76.5 | 61.2 | 40.9 | 22.4 | 35.3 |
| | SGC | 79.2 | 79.7 | 78.4 | 76.4 | 71.6 | 68.6 |
| | $S^2$GC | 78.5 | 79.2 | 79.7 | **80.2** | 79.1 | 78.1 |

Table 9: Classification accuracy (%) w.r.t. $\alpha$.

| Dataset | 0.0 | 0.05 | 0.1 | 0.15 |
|---------|-----|------|-----|------|
| Cora | 82.9 | **83.5** | 81.1 | 78.8 |
| Citeseer | 72.8 | **73.6** | 73.0 | 73.6 |
| Pubmed | 79.8 | **80.2** | 80.1 | 79.8 |

preprocessed datasets are summarized in Table 11. Table 7 shows that $S^2$GC rivals their models on 5 benchmark datasets. We provide the parameters setting in the supplementary material.

### 4.5 A DETAILED COMPARISON WITH VARIOUS NUMBERS OF LAYERS AND $\alpha$

Table 8 summaries the results for models with various numbers of layers ($K$ is the number of layers and it coincides with the number of aggregated filters in $S^2$GC). We observe that on Cora, Citeseer and Pubmed, our method consistently obtains the best performance with $K = 16$, equivalent of 16 layers. Overall, the results suggest that $S^2$GC can aggregate over larger neighborhoods better than SGC while suffering less from oversmoothing. In contrast to $S^2$GC, the performance of GCN and SGC drops rapidly as the number of layers exceeds 32 due to oversmoothing.

Table 9 summaries the results for the proposed method for various $\alpha$ ranging from 0 to 0.15. The table shows that $\alpha$ slightly improves the performance of $S^2$GC. Thus, balancing the impact of self-loop by $\alpha$ w.r.t. other filters of consecutively larger receptive fields is useful but the self-loop is not mandatory.

## 5 CONCLUSIONS

We have proposed Simple Spectral Graph Convolution ($S^2$GC), a method extending the Markov Diffusion Kernel (Section 3.2), whose feature maps emerge from the normalized Laplacian Regularization problem (Section 3.3) if $K \to \infty$. Our theoretical analysis shows that $S^2$GC obtains the right level of balance during the aggregation of consecutively larger receptive fields. We have shown there exists a connection between $S^2$GC and SGC, APPNP and JKN by analyzing spectral properties and implementation of each model. However, as our Claims I and II show that we have designed a filter with unique properties to capture a cascade of gradually increasing contexts while limiting oversmoothing by giving proportionally larger weights to the closest neighborhoods of each node. We have conducted extensive and rigorous experiments which show that $S^2$GC is competitive frequently outperforming many state-of-the-art methods on unsupervised, semi-supervised and supervised tasks given several popular dataset benchmarks.

### ACKNOWLEDGMENTS

This research is supported by an Australian Government Research Training Program (RTP) Scholarship.

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

Table 10: The statistics of datasets used for node classification and clustering.

| Dataset | # Nodes | # Edges | class | feature | Train/Dev/Test Nodes |
|---|---|---|---|---|---|
| Cora | 2, 708 | 5, 429 | 7 | 1433 | 140/500/1, 000 |
| Citeseer | 3, 327 | 4, 732 | 6 | 3703 | 120/500/1, 000 |
| Pubmed | 19, 717 | 44, 338 | 3 | 500 | 60/500/1, 000 |
| Reddit | 232, 965 | 11, 606, 919 | 41 | 602 | 152K/24K/55K |
| wiki | 2405 | 17981 | 17 | 4973 | |

Dengyong Zhou, Olivier Bousquet, Thomas Navin Lal, Jason Weston, and Bernhard Schölkopf. Learning with local and global consistency. *Advances in neural information processing systems*, 16(16):321–328, 2004.

# A   SUPPLEMENTARY MATERIAL

## A.1   NODE CLUSTERING

For $S^2GC$ and AGC, we set max iterations to $60$. For other baselines, we follow the parameter settings in the original papers. In particular, for DeepWalk, the number of random walks is 10, the number of latent dimensions for each node is 128, and the path length of each random walk is 80. For DNGR, the autoencoder is of three layers with 512 neurons and 256 neurons in the hidden layers respectively. For GAE and VGAE, we construct encoders with a 32-neuron hidden layer and a 16-neuron embedding layer, and train the encoders for 200 iterations using the Adam optimizer with learning rate equal 0.01. For ARGE and ARVGE, we construct encoders with a 32-neuron hidden layer and a 16-neuron embedding layer. The discriminators are built by two hidden layers with 16 and 64 neurons respectively. On Cora, Citeseer and Wiki, we train the autoencoder-related models of ARGE and ARVGE for 200 iterations with the Adam optimizer, with the encoder and discriminator learning rates both set as 0.001; on Pubmed, we train them for 2000 iterations with the encoder learning rate 0.001 and the discriminator learning rate 0.008.

## A.2   TEXT CLASSIFICATION

**The 20NG dataset1**   (bydate version) contains 18,846 documents evenly categorized into 20 different categories. In total, 11,314 documents are in the training set and 7,532 documents are in the test set.

**The Ohsumed corpus2**   is from the MEDLINE database, which is a bibliographic database of important medical literature maintained by the National Library of Medicine. In this work, we used the 13,929 unique cardiovascular diseases abstracts in the first 20,000 abstracts of the year 1991. Each document in the set has one or more associated categories from the 23 disease categories. As we focus on single-label text classification, the documents belonging to multiple categories are excluded so that 7,400 documents belonging to only one category remain. 3,357 documents are in the training set and 4,043 documents are in the test set.

**R52 and R8**   (all-terms version) are two subsets of the Reuters 21578 dataset. R8 has 8 categories, and was split to 5,485 training and 2,189 test documents. R52 has 52 categories, and was split to 6,532 training and 2,568 test documents.

**MR**   is a movie review dataset for binary sentiment classification, in which each review only contains one sentence (Pang & Lee, 2005) The corpus has 5,331 positive and 5,331 negative reviews. We used the training/test split in (Tang et al., 2015).

### A.2.1   TEXT CLASSIFICATION

**Parameters.**   We follow the setting of Text GCN (Yao et al., 2019) that includes experiments on four widely used benchmark corpora such as 20-Newsgroups (20NG), Ohsumed, R52 and R8 of Reuters 21578. For Text GCN, SGC, and our approach, the embedding size of the first convolution

Table 11: The statistics of datasets for text classification.

| Dataset | # Docs | # Training | # Test | # Words | # Nodes | # Classes | Average Length |
|---------|--------|-----------|--------|---------|---------|-----------|----------------|
| 20NG | 18,846 | 11,314 | 7,532 | 42,757 | 61,603 | 20 | 221.26 |
| R8 | 7,674 | 5,485 | 2,189 | 7,688 | 15,362 | 8 | 65.72 |
| R52 | 9,100 | 6,532 | 2,568 | 8,892 | 17,992 | 52 | 69.82 |
| Ohsumed | 7,400 | 3,357 | 4,043 | 14,157 | 21,557 | 23 | 135.82 |
| MR | 10,662 | 7,108 | 3,554 | 18,764 | 29,426 | 2 | 20.39 |

layer is 200 and the window size is 20. We set the learning rate to 0.02, dropout rate to 0.5 and the decay rate to 0. The 10% of training set is randomly selected for validation. Following (Kipf & Welling, 2016), we trained our method and Text GCN for a maximum of 200 epochs using the Adam (Kingma & Ba, 2014) optimizer, and we stop training if the validation loss does not decrease for 10 consecutive epochs. The text graph was built according to steps detailed in the supplementary material.

To convert text classification into the node classification on graph, there are two relationships considered when forming graphs: (i) the relation between documents and words and (ii) the connection between words. For the first type of relations, we build edges among word nodes and document nodes based on the word occurrence in documents. The weight of the edge between a document node and a word node is the Term Frequency-Inverse Document Frequency (Rajaraman & Ullman, 2011) (TF-IDF) of the word in the document applied to build the Docs-words graph. For the second type of relations, we build edges in graph among word co-occurrences across the whole corpus. To utilize the global word co-occurrence information, we use a fixed-size sliding window on all documents in the corpus to gather co-occurrence statistics. Point-wise Mutual Information (Church & Hanks, 1990) (PMI), a popular measure for word associations, is used to calculate weights between two word nodes according to the following definition:

$$\text{PMI}(i, j) = \log \frac{p(i, j)}{p(i)p(j)} \tag{14}$$

where $p(i, j) = \frac{W(i,j)}{W}$, $p(i) = \frac{W(i)}{W}$. $\#W(i)$ is the number of sliding windows in a corpus that contain word $i$, $\#W(i, j)$ is the number of sliding windows that contain both word $i$ and word $j$, and $\#W$ is the total number of sliding windows in the corpus. A positive PMI value implies a high semantic correlation of words in a corpus, while a negative PMI value indicates little or no semantic correlation in the corpus. Therefore, we only add edges between word pairs with positive PMI values:

$$\mathbf{A} = \left[ \begin{array}{c|c} \mathbf{W}_1 & \mathbf{W}_2 \\ \hline \mathbf{W}_2^\top & \mathbf{I} \end{array} \right]$$

or

$$A_{ij} = \begin{cases} \text{PMI}(i, j) & \text{if } i, j \text{ are words, PMI}(i, j) > 0, \\ \text{TF-IDF}_{ij} & \text{if } i \text{ is document, } j \text{ is word,} \\ 1 & \text{if } i = j, \\ 0 & \text{otherwise.} \end{cases} \tag{15}$$

### A.3 GRAPH CLASSIFICATION

We report the average accuracy of 10-fold cross validation on a number of common benchmark datasets, shown in Table 12, where we randomly sample a training fold to serve as a validation set. We only make use of discrete node features. In case they are not given, we use one-hot encodings of node degrees as the feature input. We note that graph classification is a task highly dependent on the global pooling strategy. There exist methods that apply sophisticated mechanisms for this step. However, with a readout function and a highly scalable $S^2GC$ model, we comfortably outperform all methods on MUTAG, Proteins and IMDB-Binary, even DiffPool which has a differentiable graph pooling module to gather information across different scales. A stronger performer (Koniusz & Zhang, 2020) uses the GIN-0 backbone and second-order pooling with the so-called spectral power normalization, referred to as MaxExp(F). In contrast, we use a simple readout feature aggregation.

Table 12: Graph classification.

| Method | MUTAG | PROTEINS | COLLAB | IMDB- BINARY |
|---|---|---|---|---|
| GCN | $74.6 \pm 7.7$ | $73.1 \pm 3.8$ | $80.6 \pm 2.1$ | $72.6 \pm 4.5$ |
| SAGE | $74.9 \pm 8.7$ | $73.8 \pm 3.6$ | $79.7 \pm 1.7$ | $72.4 \pm 3.6$ |
| GIN-0 | $85.7 \pm 7.7$ | $72.1 \pm 5.1$ | $79.3 \pm 2.7$ | $72.8 \pm 4.5$ |
| GIN-$\epsilon$ | $83.4 \pm 7.5$ | $72.6 \pm 4.9$ | $79.8 \pm 2.4$ | $72.1 \pm 5.1$ |
| DiffPool | $85.0 \pm 10.3$ | $75.1 \pm 3.5$ | $78.9 \pm 2.3$ | $72.6 \pm 3.9$ |
| GIN-0+MaxExp(F) | $88.9 \pm 5.8$ | $76.8 \pm 2.9$ | $81.7 \pm 1.7$ | $77.8 \pm 3.6$ |
| S$^2$GC | $85.1 \pm 7.4$ | $75.5 \pm 4.1$ | $80.2 \pm 1.3$ | $72.9 \pm 4.9$ |

## A.4 THEORETICAL ANALYSIS

Below we show that we can reduce oversmoothing compared to SGC while incorporating larger receptive fields.

Our design contains a sum of consecutive diffusion matrices $\widetilde{\mathbf{T}}^k, k = 0, \cdots, K$. As $k$ increases, so does the neighborhood of each node visited during diffusion $\widetilde{\mathbf{T}}^k$ (analogy to random walks). This means that:

**Claim I.** Our filter, by design, will give the highest weight to the closest neighborhood of a node as neighborhoods $\mathcal{N}$ of diffusion steps $k = 0, \cdots, K$ obey $\mathcal{N}(\widetilde{\mathbf{T}}^0) \subseteq \mathcal{N}(\widetilde{\mathbf{T}}^1) \subseteq \cdots \subseteq \mathcal{N}(\widetilde{\mathbf{T}}^K) \subseteq \mathcal{N}(\widetilde{\mathbf{T}}^\infty)$. That is, smaller neighborhoods belong to larger neighborhoods too.

To see this clearer, for the $q$-dimensional Euclidean lattice graph with infinite number of nodes, after $t$ steps of random walk, the estimate of absolute distance the walk moves from the source to its current position is given as:

$$r(t, q) = \sqrt{\frac{2t}{q}} \cdot \frac{\Gamma\left(\frac{q+1}{2}\right)}{\Gamma\left(q + 1\right)}, \tag{16}$$

where $r(t, q)$ is the absolute distance walked from the source to the current point and $\Gamma(\cdot)$ is the Gamma function. Moreover, if the number of dimensions $q \to \infty$, we have $r(t, q) \leq \sqrt{t}$. It is clear then that the receptive field associated with the random walk (and thus diffusion at time $t$) obeys the monotonically increasing radius $r$, that is $r(0) \leq r(1) \leq \cdots \leq r(K) \leq \cdots \leq r(\infty)$. To see that, simply plot $\sqrt{t}$ (and/or the more complicated expression that includes the Gamma function).

This proves Claim I for the Euclidean lattice graph. That is, for consecutive diffusion steps $\widetilde{\mathbf{T}}^k, k = 0, \cdots, K$, our receptive field grows.

Moreover, note that our filter is realized as the sum of consecutive diffusion steps, that is $\frac{1}{t} \sum_{\tau=0}^{t} \text{diff}(s, \tau)$ where $s$ is the source of walk. It is easy to see then that even if each walked distance was to contribute the energy proportional with $r(t)$ to the summation term, we have:

$$\lim_{t \to \infty} \frac{\frac{1}{t} \sum_{t'=0}^{t} \sqrt{t'}}{\sqrt{t}} = 0, \tag{17}$$

where the enumerator is the model of the total energy when aggregating over receptive fields from size 0 to $\infty$ in S$^2$GC while the denominator is the total energy of SGC (filter is given by $\widetilde{\mathbf{T}}^K$, that is by $\text{diff}(s, t)$).

The above proof shows that the above ratio of energies is 0, which means that:

**Claim II.** When the ratio of energies of two models is 0, the energy of the infinite-dimensional receptive field (when $t \to \infty$) in S$^2$GC is not going to dominate the sum energy of our filter. Thus, S$^2$GC can incorporate larger receptive fields than SGC without eclipsing the contributions from smaller receptive fields as $t \to \infty$ on the Euclidean lattice graph.

However, in practice, we work with finite-dimensional non-Euclidean graphs. Obtaining the absolute distance $r(t)$ walked from the source is a difficult topic. As an example, see Eq. 184 in Masuda et al. (2017).

For this reason, below we use a simple approximation. We use Theorem 1 as the proxy for the walked radius. That is to say the error of convergence to the stationary distribution is indicative of the absolute distance walked from the source/node. Specifically, we have:

Recall Theorem 1, that is let $\lambda_2$ denote second largest eigenvalue of transition matrix $\widetilde{\mathbf{T}} = \mathbf{D}^{-1}\mathbf{A}$, $\mathbf{p}(t)$ be the probability distribution vector and $\boldsymbol{\pi}$ the stationary distribution. If walk starts from the vertex $i$, $p_i(0) = 1$, then after $t$ steps for every vertex:

$$|p_j(t) - \pi_j| \leq \sqrt{\frac{d_j}{d_i}} \lambda_2^t. \tag{18}$$

Then, the average walked distance $r$ from node $i$ over $t$ steps in a graph with $n$ nodes and connectivity given by the second largest eigenvalue $\lambda_2$, denoted by $r(i,t,n)$ is lower-bounded by $\bar{r}(i,t,n)$ as follows:

$$r(i,t,n) \approx \frac{1}{\frac{1}{n-1}\sum\limits_{j \neq i}|p_j(t) - \Pi_j|} \geq \bar{r}(i,t,n) = \frac{n-1}{\lambda_2^t \frac{\sum_{j \neq i}\sqrt{d_j}}{\sqrt{d_i}}} = \frac{(n-1)\sqrt{d_i}}{\lambda_2^t(\widetilde{E}-\sqrt{d_i})} = \frac{\rho}{\lambda_2^t}, \tag{19}$$

where $n$ is the number of nodes, $t$ is the number of diffusion steps (think $\widetilde{\mathbf{T}}^k$), $d_i$ and $d_j$ are degrees of nodes $i$ and $j$, $\lambda_2$ being the second largest eigenvalue intuitively denotes the graph connectivity (large $\lambda_2 \leq 1$ indicates low connectivity while low $\lambda_2$ indicates high connectivity in graph), $\widetilde{E}$ is the sum of square roots of node degrees and $\rho = \frac{(n-1)\sqrt{d_i}}{\widetilde{E}-\sqrt{d_i}}$ is in fact a constant for a given graph.

While the above approximations may be loose for very small/large $t$, the important property to note is that $r(i,0,n) \leq r(i,1,n) \leq \cdots \leq r(i,t,n)$ which indicates that our filter indeed realises the sum over increasingly larger receptive fields. As smaller receptive fields are a subset of larger receptive fields given node $i$, that is $\mathcal{N}(\widetilde{\mathbf{T}}^0) \subseteq \mathcal{N}(\widetilde{\mathbf{T}}^1) \subseteq \cdots \subseteq \mathcal{N}(\widetilde{\mathbf{T}}^K) \subseteq \mathcal{N}(\widetilde{\mathbf{T}}^\infty)$, this proves our Claim I.

To prove Claim II for a general connected non-bipartite graph, we have:

$$\lim_{t \to \infty} \frac{\frac{1}{t}\sum\limits_{t'=0}^{t} \bar{r}(i,t',n)}{\bar{r}(i,t,n)} = 0, \tag{20}$$

Similar findings can be noted by carefully considering the meaning of so-called Cheeger constant introduced in Section A.5. More details on spectral analysis of filters in GCNs can be found in studies of Li et al. (2018a) and Li et al. (2018b).

## A.5  GRAPH PARTITIONING

Below we introduce the definitions of expansion and $k$-way Cheeger constant.

**Definition A.1.** For a node subset $S \subseteq V$, so-called expansion $\phi(S) = \frac{|\mathrm{E}(S)|}{\min\{\mathrm{vol}(S), \mathrm{vol}(V \setminus S)\}}$, where $\mathrm{E}(\mathbf{S})$ is the set of edges with one node in $\mathbf{S}$ and $\mathrm{vol}(\mathbf{S})$ is the sum of degree of nodes in set $\mathbf{S}$.

**Definition A.2.** The $k$-way Cheeger constant is given as: $\rho_G(k) = \min_{S_1, S_2, \cdots, S_k} \max\{\phi(S_i) : i = \{1, \cdots, k\}\}$ where the minimum is over all collections of $k$ non-empty disjoint subsets $S_1, S_2, \cdots, S_k \subseteq V$.

According to the definitions, the expansion in Def. A.1 describes the effect of graph partitioning according to subset $S$ while the $k$-way Cheeger constant reflects the effect of the graph partitioning into $k$ parts–the smaller the value the better the partitioning is. Higher-order Cheeger's inequality (Bandeira et al., 2013; Lee et al., 2014) bridges the gap between the network spectral analysis and graph partitioning by controlling the bounds of $k$-way Cheeger constant:

$$\frac{\lambda_k}{2} \leq \rho_G(k) \leq \mathcal{O}\left(k^2\right)\sqrt{\lambda_k}, \tag{21}$$

where $\lambda_k$ is the k-th eigenvalue of the normalized Laplacian matrix and $0 = \lambda_1 \leq \lambda_2 \leq \cdots \leq \lambda_n$. From inequality 21, we can conclude that small (large) eigenvalues control global clustering (local smoothing) effect of the graph partitioned into a few large parts (many small parts). Thus, specific combination of low- and high-pass filtering of our design (see Figure 1) a indicates the weight trade-off between large and small partitions contained by the node.

