# OpenReview forum: "Simple Spectral Graph Convolution"
_ICLR.cc/2021/Conference — ICLR 2021 Poster_

### Official Review · AnonReviewer2 · 2020-10-27
**Review for "Simple Spectral Graph Convolution"**

**Rating:** 7
**Confidence:** 4

**Review:**

**Summar**
The paper proposes a spectral-based graph convolution layer, called Simple Spectral Graph Convolution (S$^2$GC), which is based on the Markov Diffusion Kernel (MDK). The authors show that S$^2$GC is capable of aggregating k-hop neighbourhood information without oversmoothing. The paper provides a spectral analysis on S$^2$GC and shows the connections to several previous methods, such as GDC, SGC and APPNP. The authors also show that their proposed S$^2$GC advantages from both spatial and spectral methods. They demonstrate that their S$^2$GC on a series of experiments, such as node clustering and node classification.

**Reasons for score**
Overall, I recommend ``marginally above the acceptance threshold''. It is plausible the idea of using Markov Diffusion Kernel (MDK) to derive a spectral graph convolution which can effectively aggregate k-hop information, and at the same time prevent from oversmoothing due to the design of MDK. The primary concern is about the clarity of the paper on the explanation of the required computational and storage costs of the proposed S$^2$GC model. It is suggested that the authors can address it in the rebuttal period. The following are the detailed comments.

**Pros**
1. The paper takes one of the most important issues of graph convolution: oversmoothing when aggregating multi-hop neighbourhoods / using deep graph neural networks. The problem itself is practical and makes much sense.
2. The paper provides a rigorous and detailed spectral analysis for their proposed S$^2$GC method and shows the connection and difference between S$^2$GC and some previous related spectral graph convolutions. Overall, the paper is well-written.
3. The experiments provided in this paper are extensive and quite comprehensive. Specifically, the authors demonstrate the effectiveness and superiority of their proposed S$^2$GC through four different tasks. The experimental results are also convincing.

**Cons**
1. The paper provides an experiment (community prediction) on a large-scale dataset. I will suggest the authors use some large-scale datasets also for the semi-supervised node classification task, such as Open Graph Benchmark (OGB) --- Node Property Prediction dataset, in addition to the over-used classic citation networks.
2. It would be more interesting if the authors could provide a strategy on choosing the optimal Step Parameter $K$ in practice. Alternatively, the authors can provide an experiment on how the performance of S$^2$GC model is influenced by different values of the Step Parameter $K$.
3. It would be even better if the authors can elaborate the computational and storage costs of their S$^2$GC method compared to the previously mentioned methods in the paper, such as GDC, SGC and APPNP.

Questions during the rebuttal period: Please address and clarify the Cons above.

**Typos**
1. In the paragraph of Section 2 Preliminaries: Is the paragraph title "APPNA" supposed to be "APPNP"?
2. In Section 4 Experiments: I would suggest swapping the order of Table 1 and Table 2 since Table 2 is mentioned before Table 1 in the text.

---

### Official Review · AnonReviewer3 · 2020-10-28
**New Graph Filter with Strong Empirical Performance**

**Rating:** 6
**Confidence:** 4

**Review:**

Summary:
One of the most important component of GCN is coming with suitable graph filters and crucial towards designing better GCNs. In this regard, authors proposed Simple Spectral Graph Convolution with Markov diffusion kernel as a graph filter which combines strengths of both spatial and spectral methods. The paper is written-well and easy to follow.  Simplicity and empirically strong results are the main contributions of the paper. However, experiments on graph classification, comparison with other graph filters and theoretical justification can help make the stronger case.


Pros:
Paper proposes Markov diffusion kernel as the new filter for GCN which can be seen as a variant (or combo) of Simple Graph Convolution and  APPNP. The main Equation 13 solves both the issues of propagating information to higher depth i.e., oversmoothing and preserving the node self information. Although, both these two tricks are somewhat known but showing their dominance in the empirical performance is promising and novel to certain extent.


Cons:
Theoretical analysis is quite limited. Note that a similar Theorem exits in (Li et al., 2018)[1] to show over smoothing issue in GCNs. A comparison or a contrast with Theorem 1 would be beneficial. Also paper lacks insights about GCNs on how to design better graph filters and comparison with other graph filters (see [2]).  Moreover, authors are missing graph classification task which is quite common to show the efficacy of a new graph filter and can help to make stronger case.


Reproducibility:
Paper lacks the implementation details and it would be great if authors can open-source their code for reproducibility purposes. I am suspicious about the standard deviation being zero in Table 4 on Cora and Pubmed datasets suggesting that the either algorithm is too stable or experiments lacks the needed randomness for statistically significant results.



[1] Qimai Li, Zhichao Han, and Xiao-Ming Wu. Deeper insights into graph convolutional networks for
semi-supervised learning. In Proceedings of the Thirty-Second AAAI Conference on Artificial In-
telligence (AAAI-18), pp. 3538–3545. Association for the Advancement of Artificial Intelligence,
2018.

[2] Adaptive Graph Convolutional Neural Networks Ruoyu Li, Sheng Wang, Feiyun Zhu, Junzhou Huang

---

### Official Review · AnonReviewer1 · 2020-10-29
**The paper proposes a relatively simple formulation for a graph convolution based on a K-step Markov Diffusion kernel. The novelty of the proposed approach should be better detailed, and experimental results should be improved.**

**Rating:** 6
**Confidence:** 4

**Review:**

The paper proposes a relatively simple formulation for a graph convolution that considers nodes at multiple distances from the root node.
The novelty of the proposed approach should be better detailed, and experimental results should be improved.

The proposed formulation in eq. 13 is uncannily similar to the one proposed by Defferrard et al https://arxiv.org/pdf/1606.09375.pdf known as Chebynet, with a different choice of polynomials. Authors also acknowledge their proposal is a special case of GDC.
Author then follow the same principle as PPNP to incorporate information about node labels at each layer. These points pose an issue about the novelty of the proposed approach, that should be better discussed.

Pros:
-The proposed method is relatively simple and shows good performance
-In the conclusions and in the introduction, authors claim that their proposed approach avoids oversmoothing. However, this is not proven in the paper. If authors refer to the introduction of the term \alpha X in the convolution, this was proposed by PPNP already, so it is not a contribution of the present paper. Author should discuss this point more in the manuscript.

Cons:
Experimental results can be improved.
-Table 1 does not report variance. Moreover, the proposed methods' performance are very close to the ones of SGC. Table 2 does not report variance.
-How were the hyper-parameters K, \alpha and the learning rate set for the different datasets? At least a study reporting the influence on the predictive performance of different choices should be provided. Even better, the hyper-parameters should be selected using a validation set. For more details about the importance of the evaluation procedure, please see https://openreview.net/pdf?id=HygDF6NFPB

Minor:
-The tables are not reported in the order they are mentioned in the text.
-Page 1 "the features of on the Euclidean" -> features on Euclidean grids
-Page 3 : APNNA -> APNNP?
-Table 6: missing bold for SGC on Ohsumed dataset. S^2GC variance of 0.0 on MR looks strange.


-----AFTER REBUTTAL
I have carefully checked the author rebuttal, and it addressed several of my concerns. I thus improve my score to this paper.

---

### Official Review · AnonReviewer4 · 2020-10-30
**Novelty is limited**

**Rating:** 5
**Confidence:** 4

**Review:**

This paper proposes a graph convolutional network, S2GC. The major difference between S2GC and previous graph convolutional networks is the proposed “new” graph convolution filter. This paper also systematically analyses previous methods and discusses their relations with S2GC.

This paper gives a comprehensive review and discussion about preliminaries works. The authors also conduct extensive experiments on node clustering, community prediction, node classification, and text classification.

My major concerns are:

1)	The major difference between S2GC and previous methods, such as GCN and GDC (eq 4), is the graph filter. However, the proposed filter is not a new one. As in discussed in sec 3.3, it is the solver of normalized Laplacian regularization problem (eq. 14), which has been well studied in the past few decades. For example, the AR filter in [a] is the same as the one in S2GC. [a] also provided detailed spectral analysis as this paper does. The difference is that [a] only used this filter for node classification, while this paper also tests on node clustering, community prediction and text classification.

2)	While the authors conduct many experiments, the proposed filter does not bring significant performance improvement, which, however, is consistent with the observations in [a]. In most cases, the improvements are marginal, if any.

[a] Label Efficient Semi-Supervised Learning via Graph Filtering, CVPR2019

=================== Post Rebuttal ======================================

I would like to thank the authors for the feedback.
I realize that the diffusion kernel used in this paper is different from the AR filter in the CVPR 2019 paper, but I was misled to think they are the same because of a wrong claim made by the authors. In their paper, right below Eq. (11) (of the latest version), the authors state that the limiting case of the diffusion kernel is the solution of the classical Laplacian regularization problem, which is not true. The solution of this problem is exactly the AR filter with alpha = 1, and it is not equivalent to the limiting case of the diffusion kernel.

As such, I increased my rating of the paper, but I still think the contribution of the paper is limited due to the following reasons.

1.	There is not much novelty in using the diffusion kernel as the graph filter. The diffusion kernel has similar response function as the AR filter. If you plot the response functions and observe the functions within [-1, 1] (range of the eigenvalues), you may see they nearly overlap. Yes, the diffusion kernel filter let more high frequency signals pass, but the amount is quite small, and it does not make much difference. The theoretical claims are not insightful either. Claim 1 is obvious, and the conclusion of Claim 2 applies to other polynomial filters such as AR (AR can be expanded into a polynomial form, the CVPR 2019 paper also proposed a fast computation method similar to what used in this paper).

2.	The performance of the diffusion kernel is not significantly better than other graph filters. Notice that, it is not fair to directly compare with the results in the CVPR 2019 paper, because those results were obtained without using the validation set (which normally contains 500 labeled examples as in the original GCN paper). Also, the results in the present paper are obtained by running through a wide range of k and selecting the best. Given same conditions, it is quite likely other graph filters can achieve the same performance. Also, some recent methods such as GMNN and GCNII achieved better results than the present paper.

I hope the authors don’t think they are treated by harsher standards. The novelty of the CVPR 2019 work is not mainly about proposing a new filter, it is about providing insights into/unifying popular semi-supervised learning frameworks such as label propagation and GCN. Especially for GCN, there was no similar analysis on GCN two years ago (an early version of the CVPR 2019 paper is online (https://openreview.net/forum?id=SygjB3AcYX) months before the SGC paper).

---

### Decision · Program_Chairs · 2021-01-07
**Final Decision**

**Decision:**

Accept (Poster)

**Comment:**

The authors propose a Simple Spectral Graph Convolution (S2GC) variant of GCNs, using a modified Markov Diffusion kernel. The approach proposed aims to tackle GCN performance degradation with increased depth (oversmoothing), by combining techniques from earlier works (Simple Graph Convolution and APPNP). This is complemented by a spectral analysis of the properties of the scheme, as well by a comparison to state-of-the art on several datasets.